# Luteolin-7-O-Glucoside Inhibits Oral Cancer Cell Migration and Invasion by Regulating Matrix Metalloproteinase-2 Expression and Extracellular Signal-Regulated Kinase Pathway

**DOI:** 10.3390/biom10040502

**Published:** 2020-03-26

**Authors:** Bharath Kumar Velmurugan, Jen-Tsun Lin, B. Mahalakshmi, Yi-Ching Chuang, Chia-Chieh Lin, Yu-Sheng Lo, Ming-Ju Hsieh, Mu-Kuan Chen

**Affiliations:** 1Toxicology and Biomedicine Research Group, Faculty of Applied Sciences, Ton Duc Thang University, Ho Chi Minh City 700000, Vietnam; bharath.kumar.velmurugan@tdtu.edu.vn; 2Division of Hematology and Oncology, Department of Medicine, Changhua Christian Hospital, Changhua 500, Taiwan; 111227@cch.org.tw; 3School of Medicine, Chung Shan Medical University, Taichung 402, Taiwan; 4Institute of Research and Development, Duy Tan University, Da Nang 550000, Vietnam; mahalakshmibharath05@gmail.com; 5Oral Cancer Research Center, Changhua Christian Hospital, Changhua 500, Taiwan; 177267@cch.org.tw (Y.-C.C.); 181327@cch.org.tw (C.-C.L.); 165304@cch.org.tw (Y.-S.L.); 6Institute of Medicine, Chung Shan Medical University, Taichung 402, Taiwan; 7Department of Holistic Wellness, MingDao University, Changhua 52345, Taiwan; 8Graduate Institute of Biomedical Sciences, China Medical University, Taichung 404, Taiwan; 9Department of Otorhinolaryngology, Head and Neck Surgery, Changhua Christian Hospital, Changhua 500, Taiwan

**Keywords:** luteolin-7-O-glucoside, oral cancer, migration, invasion, MMP-2

## Abstract

Oral squamous cell carcinoma is the sixth most common type of cancer globally, which is associated with high rates of cancer-related deaths. Metastasis to distant organs is the main reason behind worst prognostic outcome of oral cancer. In the present study, we aimed at evaluating the effects of a natural plant flavonoid, luteolin-7-O-glucoside, on oral cancer cell migration and invasion. The study findings showed that in addition to preventing cell proliferation, luteolin-7-O-glucoside caused a significant reduction in oral cancer cell migration and invasion. Mechanistically, luteolin-7-O-glucoside caused a reduction in cancer metastasis by reducing p38 phosphorylation and downregulating matrix metalloproteinase (MMP)-2 expression. Using a p38 inhibitor, SB203580, we proved that luteolin-7-O-glucoside exerts anti-migratory effects by suppressing p38-mediated increased expression of MMP-2. This is the first study to demonstrate the luteolin-7-O-glucoside inhibits cell migration and invasion by regulating MMP-2 expression and extracellular signal-regulated kinase pathway in human oral cancer cell. The study identifies luteolin-7-O-glucoside as a potential anti-cancer candidate that can be utilized clinically for improving oral cancer prognosis.

## 1. Introduction

Oral squamous cell carcinoma that primarily affects the lips, oral cavity, and tongue represents about 90% of all head and neck cancers. Together with pharyngeal cancer, oral carcinoma has become the sixth most common type of cancer globally [1]. Despite advancement in diagnostic processes and therapeutic interventions, oral cancer is associated with a high rate of morbidity and mortality [2]. As a result of the initial asymptomatic nature, oral cancer is often diagnosed in the later stage, resulting in distant metastasis and poor prognosis [3,4,5,6].

Metastasis that occurs in the advanced stage of cancer is the leading cause of cancer-related deaths [5]. The extra cellular matrix degradation by proteases, including matrix metalloproteinase (MMPs), is a vital phenomenon associated with cancer cell migration [7]. An overexpression or increased activity of MMPs is associated with higher cancer aggressiveness and poor survival rate [8]. MMPs are a target for developing treatment strategies against cancer. In this context, several studies have shown that selective inhibition of MMP can be associated with better cancer management [9,10,11].

Mitogen-activated protein kinases (MAPKs), including p38 MAP kinase, Jun N-terminal kinase 1/2/3 (JNK1/2/3), and extracellular signal-regulated kinase 1/2 (ERK1/2), are the primary regulators of proliferation, differentiation, apoptosis, migration, and invasion of cancer cells [12]. Introducing a gap between MAPK signaling pathway and MMP activity, it is known that the stimulation of p38 causes increased metastasis of neoplastic squamous epithelial cells by regulating the expression of MMPs [13]. In prostate cancer, p38 MAPK is known to trigger invasion by increasing MMP-2 expression and activity [14]. Moreover, breast cancer cells lacking p38 MAPK expression have been shown to have reduced MMP-9 activity and significantly lower rate of bone metastasis [15]. Interestingly, ERK1/2 and p38 MAPK maintains an inverse relationship wherein a high ERK activity and a low p38 activity result in increased cell growth and survival [16]. Regarding cancer metastasis, it has been found that stimulation of ERK phosphorylation leads to increased cancer metastasis [17,18,19,20]. Moreover, previous studies have shown that human lung cancer cell migration and invasion can be suppressed by pharmacologically downregulated ERK/p38 signaling pathway and inhibited MMP-2 and MMP-9 activities [21]. Similarly, in ovarian cancer, increased cell growth and migration by Rap1A, a Ras-associated protein, has been shown to be associated with elevated ERK/p38 and notch signaling [22]. Taken together, it is well-evidenced that the complex crosstalk between MAPK signaling pathway components and MMPs plays an immensely important role in regulating cancer metastasis and progression. Thus, selective targeting of any of these components through pharmacological interventions can be an effective strategy to ameliorate cancer-related burdens, especially metastasis.

Luteolin (3’,4’,5,7-hydroxyl-flavone) is a naturally occurring plant flavonoid that has been used extensively in Chinese traditional medicine because of several health benefits, including anti-inflammatory, antioxidative, and anti-cancer effects [23,24]. Mostly, luteolin is present in plants as a glycosylated component (glucoside), which is hydrolyzed in the gut to produce free luteolin during absorption [25]. Several studies evaluating the therapeutic properties of luteolin have potentiated its anti-cancer effects, which are primarily associated with increased cancer cell death, reduced proliferation, and angiogenesis, and increased cancer cell sensitization to chemotherapies [23,24,26]. Moreover, luteolin has been shown to have preventive effects against cytotoxicity produced by chemotherapeutic agents, such as cisplatin [27].

Despite having a large number of evidences on chemopreventive and anti-proliferative properties of luteolin, its effect on cancer metastasis has not been studied extensively. In the present study, we aimed at evaluating the effects of luteolin-7-O-glucoside on oral cancer cell migration and invasion. A detailed mechanism of action of luteolin-7-O-glucoside was also studied.

## 2. Materials and Methods

### 2.1. Cell Culture

Human oral squamous carcinoma cell lines (FaDu and HSC-3) were received from ATCC (Manas, VA, USA). In addition, Ca9-22 cell line was obtained from Japanese Collection of Research Bioresources Cell Bank (JCRB, Shinjuku, Japan) and cultured in the in Dulbecco’s Modified Eagle Medium (DMEM; Life Technologies, Grand Island, NY, USA): Ham’s F12 Nutrient Mixture (Life Technologies, Grand Island, NY, USA) supplemented with 10% FBS (Invitrogen, Waltham, MA, USA) [28]. The cells were cultured in 5% CO_2_ at 37 °C.

### 2.2. Luteolin-7-O-Glucoside Treatments

Luteolin-7-O-glucoside (≥97% purity) was obtained from Sigma-Aldrich (St. Louis, MO, USA). Luteolin-7-O-glucoside stock solution (100 mM) was prepared using dimethyl sulfoxide (DMSO) and stored at −20 °C. The DMSO concentration was less than 0.2% for each experiment. For luteolin-7-O-glucoside treatments, appropriate amounts of stock solution were administered to the medium to get the final experimental doses.

### 2.3. MTT Assay

To study the effects of luteolin-7-O-glucoside on cell viability, MTT (3-(4,5-dimethylthiazol-2-yl)-2,5-diphenyltetrazolium bromide) assay was conducted using HSC-3, FaDu, and CA9-22 cells. The cells were seeded onto 24-well plates and treated with 0 to 40 μM of luteolin-7-O-glucoside solutions at 37 °C for 24 h. Cell viability was determined following methods described previously [29].

### 2.4. Wound Healing Assay

Oral cancer cells FaDu, CA9-22, and HSC-3 were seeded into 12-well plates and cultured to 90% confluence. A cell monolayer was scratched with a 1-mL micropipette tip in each well, and treated luteolin-7-O-glucoside (0, 10, 20, and 40 μM) at 37 °C for 0, 2, 4, 6, 24 h. The cells were photographed under a microscope for its migration ability and its mean crawling distance of cells was measured.

### 2.5. Cell Migration and Invasion Assay

Migration and invasion assays were performed as described previously [28,30]. Oral cancer cells were mixed in serum free medium and seeded into the upper chambers of the inserts (Greiner Bio-One, Monroe, NC, USA). The inserts were placed in 24-well plates containing complete medium with various concentration of luteolin-7-O-glucoside in lower wells. Migration and invasion ability was observed and captured under light microscope.

### 2.6. Western Blot Assay

Western blot analysis was performed as previously described [31]. The membranes were blocked with 5% non-fat milk in TBST (Tris-Buffered Saline, 0.1% Tween 20 Detergent) for 1 h and incubated with indicated primary antibodies for 24 h at 4 °C. After washing, the blots were incubated with peroxidase-conjugated secondary antibodies for 1 h. Finally, bands were monitored using were visualized through an ECL detection system.

### 2.7. Statistical Analysis

All experiments were repeated at least 3 times. All statistical analysis was performed by Student’s *t*-test (Sigma Plot, version 10.0). *p* < 0.05 is considered as statistically significant.

## 3. Results

### 3.1. Luteolin-7-O-Glucoside Exerts Cytotoxic Effects on Human Oral Cancer Cells

To investigate whether luteolin-7-O-glucoside can alter cell proliferation, three oral cancer cell lines, FaDu, HSC-3, and CA9-22, were treated with various concentrations (0, 10, 20, and 40 μM) of luteolin-7-O-glucoside for 24 h, and the cell viability was determined by MTT assay. As observed in Figure 1a–c, the treatment with 20 and 40 μM of luteolin-7-O-glucoside significantly reduced the viability of oral cancer cells as compared to untreated controls, indicating the growth inhibition potential of the compound.

### 3.2. Luteolin-7-O-Glucoside Inhibits Motility of Human Oral Cancer Cells

In addition to cell proliferation, we investigated the effects of luteolin-7-O-glucoside on cell motility. As mentioned previously, FaDu, HSC-3, and CA9-22 cells were treated with various concentrations (0, 10, 20, and 40 μM) of luteolin-7-O-glucoside for 3, 6, and 24 h. The analysis of cell motility using wound closer assay revealed that luteolin-7-O-glucoside significantly reduced the cancer cell migration in a dose-dependent manner (Figure 2a–f). These findings indicate that luteolin-7-O-glucoside possesses anti-migratory activity.

### 3.3. Luteolin-7-O-Glucoside Inhibits Migration and Invasion of Human Oral Cancer Cells

To investigate whether luteolin-7-O-glucoside can inhibit both migration and invasion, FaDu, HSC-3, and CA9-22 cells were treated similarly as the previous experiments. As observed in Figure 3a,b, luteolin-7-O-glucoside significantly reduced the migration of all three oral cancer cells in a dose-dependent manner. Similarly, a significant reduction in invasion was observed after 24 h treatment with luteolin-7-O-glucoside (Figure 3c,d). Taken together, all these findings clearly indicate that luteolin-7-O-glucoside is capable of significantly ameliorating the metastatic profile of oral cancer cells.

### 3.4. Luteolin-7-O-Glucoside Reduces the Protein Expression of MMP-2 in Human Oral Cancer Cells

Next, we investigated the molecular mechanism responsible for the luteolin-7-O-glucoside activity. Given the significant involvement of MMPs in cancer cell migration and invasion, we checked the protein expression of MMP-2 in FaDu and HSC-3 cells after the treatment with various concentrations (0, 10, 20, and 40 μM) of luteolin-7-O-glucoside for 24 h. As observed in Western blot analysis, all the doses of luteolin-7-O-glucoside significantly reduced the expression of MMP-2 in both the cell lines (Figure 4a–d). These findings indicate that luteolin-7-O-glucoside reduces oral cancer cell migration and invasion by decreasing the cellular level of MMP-2.

### 3.5. Luteolin-7-O-Glucoside Inhibits the p38 Pathway in Human Oral Cancer Cells

Since the expressions and activities of MMPs are tightly regulated by MAPK pathway components, we next investigated the protein phosphorylation status of ERK, p38, and JNK in FaDu and HSC-3 cells after employing similar treatment as before. As observed in Figure 5a,b, luteolin-7-O-glucoside treatment caused a significantly reduction in p38 and JNK phosphorylation in HSC-3 cells. In contrast, the phosphorylation of ERK increased significantly after the treatment. In case of FaDu cells, the treatment caused similar reduction in p38 phosphorylation as observed in HSC-3 cells (Figure 5c,d). However, the phosphorylation of JNK increased significantly after the treatment. Moreover, the phosphorylation of ERK reduced significantly only after the treatment with 40 μM of luteolin-7-O-glucoside (Figure 5c,d).

### 3.6. Effect of SB203580 and Luteolin-7-O-Glucoside Co-Treatment on MMP-2 Protein Expression in Human Oral Cancer Cells

Since we observed a consistent change in p38 phosphorylation status in all the oral cancer cells, we next thought of using a p38 inhibitor, SB203580, to further evaluate the mechanistic details of luteolin-7-O-glucoside action. Both FaDu and HSC-3 cells were pretreated with SB203580 for 1 h, followed by treatment with 40 μM of luteolin-7-O-glucoside for 24 h. As observed in Figure 6a–d, the collective effects of SB203580 and luteolin-7-O-glucoside caused a further reduction in MMP-2 expression, indicating that luteolin-7-O-glucoside mediates anti-migratory effects by altering p38-induced activation of MMP-2.

### 3.7. Effect of SB203580 and Luteolin-7-O-Glucoside Co-Treatment on Cell Motility in Human Oral Cancer Cells

To check the effects of SB203580 and luteolin-7-O-glucoside co-treatment on cell motility, we performed wound closer assay in FaDu and HSC-3 cells using the similar treatment as mentioned before. As observed in Figure 7a–d, the co-treatment caused further reduction in cell motility as compared to the luteolin-7-O-glucoside treatment alone. These findings further confirm the involvement of p38 pathway in mediating the effects of luteolin-7-O-glucoside in oral cancer cells.

### 3.8. Effect of SB203580 and Luteolin-7-O-Glucoside Co-Treatment on Cell Migration in Human Oral Cancer Cells

Given the significant effect of SB203580 and luteolin-7-O-glucoside co-treatment on cell motility, we performed trans-well migration assay to finally prove the involvement of p38 pathway and MMP-2 in mediating luteolin-7-O-glucoside-induced reduction in oral cancer cell migration. As clearly observed in Figure 8a–c, the co-treatment resulted in a significantly higher reduction in cell migration as compared to the luteolin-7-O-glucoside treatment alone. Taken together, all these findings clearly indicate that luteolin-7-O-glucoside-induced inhibition of oral cancer cell metastasis is mediated by the combined action of p38 and MMP-2.

## 4. Discussion

In this study, we evaluated the anti-migratory and anti-invasive effects of luteolin-7-O-glucoside, a natural flavonoid, on oral cancer cells. To the best of our knowledge, this is the first study describing the potential role of luteolin-7-O-glucoside as an anti-metastatic agent. The findings of the study can open up a new path toward developing novel therapeutic candidates for preventing distant metastasis of oral cancer, which is known to be responsible for higher rates of cancer-related deaths worldwide.

The study was designed to evaluate the effects of different doses of luteolin-7-O-glucoside (10, 20, and 40 μM) on oral cancer cells, as well as to compare the findings with untreated controls. Of these doses, 20 and 40 μM have shown that luteolin-7-O-glucoside significant anti-proliferative or cytotoxic effects (Figure 1). These findings are in line with previous studies showing the pro-apoptotic effects of luteolin-7-O-glucoside on cancer cells [27,32,33]. A study using human liver cancer cells has shown thatluteolin-7-O-glucoside causes cancer cell death by inducing cell cycle arrest at G2/M phase, and the effect is mediated by increased production of free radical and phosphorylation of JNK [34]. Moreover, a recent study has shown that luteolin-7-O-glucoside extracted from *Cuminum cyminum* has cytotoxic effects against breast cancer cells, which makes luteolin-7-O-glucoside a potential chemotherapeutic agent [35]. In contrast, it has been found that luteolin-7-O-glucoside exerts anti-apoptotic effects on cardiomyocytes that are treated with angiotensin II to develop cardiac hypertrophy [36]. The cardio-protective effects of luteolin-7-O-glucoside has been found to be associated with reduced free radical level, improved antioxidative capacity, and decreased hypoxia/reperfusion-induced cell death. Luteolin-7-O-glucoside has been found to mediate all these effects by modulating the MAPK signaling pathway. These findings clearly suggest that luteolin-7-O-glucosideexerts cytoprotective effects on normal, non-cancerous cells, such as cardiomyocytes.

Regarding the impact of luteolin-7-O-glucoside on cancer cell motility, we found that the compound is capable of significantly reducing the migration and invasion of oral cancer cells in a dose-dependent manner (Figure 2 and Figure 3). Although there is no direct evidence on the anti-migratory activity of luteolin-7-O-glucoside, luteolin has been shown to reduce migration by inhibiting the production and secretion of pro-inflammatory cytokines, such as TNFα and IL-6 [37], which are known to induce cell migration by increasing the expression of migration-promoting proteins, including MMPs [38]. Moreover, luteolin has been found to inhibit the migration and invasion of pancreatic cancer cells by trans-inactivating EGFR activity and suppressing the phosphorylation of focal adhesion kinase (FAK) and MMP secretion [39]. In case of liver cancer cells, luteolin has been found to inhibit cell migration and invasion by reducing the phosphorylation of c-Met (hepatocyte growth factor receptor), as well as modulating the MAPK and PI3K-AKT pathways [40]. A recent study using human squamous carcinoma cell lines has shown that luteolin ameliorates cancer metastasis by reducing the expression and activity of ribosomal protein S19 and inhibiting the AKT/mTOR/c-Myc signaling pathway [41]. Luteolin suppress vascular endothelial growth factor receptor (VEGF) 2 and mediated human prostate tumor growth and angiogenesis [42]. In addition, the migration of squamous carcinoma cells has been found to be modulated by luteolin-induced reduction in expressions and activities of S100A7, Src, and STAT3 [43].

In addition, we also investigated the mode of action of luteolin-7-O-glucoside. According to the results in Figure 5, luteolin-7-O-glucoside treatment caused a significantly reduction in p38 and JNK phosphorylation in HSC-3 cells. In case of FaDu cells, the treatment caused similar reduction in p38 phosphorylation as observed in HSC-3 cells (Figure 5c,d). The study findings demonstrated that luteolin-7-O-glucoside mediates anti-metastatic effects by reducing MMP-2 expression and p38 phosphorylation (Figure 4 and Figure 5). Using a p38 inhibitor, SB203580, we confirmed that luteolin-7-O-glucoside reduces oral cancer cell migration by altering p38-induced activation of MMP-2 (Figure 6, Figure 7, and Figure 8). The anti-migratory effects of natural flavonoids such as luteolin has been well-documented in the literature [44,45,46,47]. Moreover, previous studies have shown that anti-metastatic effects of natural bioactive compounds are primarily mediated by modulation of MAPK pathways and MMP expression [48,49,50].

Several MMPs such as MT1-MMP, MMP-2, and MMP-9 were found correlated to the stages of cancer cell invasion progression. Specifically, our study findings are in line with a recent study showing that two bioactive compounds of pomegranate namely ellagic acid and luteolin prevent proliferation and migration of ovarian cancer cells by reducing the expression of MMP-2 and MMP-9 [51]. Similarly, luteolin has been found to reduce the proliferation, migration, and invasion of human melanoma cells by suppressing the expression of MMP-2 and MMP-9 and altering the PI3K/AKT signaling pathway [52]. Regarding the effect of luteolin on the p38 signaling pathway, a recent study using gastric cancer cells has shown that luteolin exerts anti-proliferative and anti-migratory effects by inhibiting the phosphorylation of PI3K, AKT, mTOR, ERK, and p38, as well as reducing the expression of MMP-2 and MMP-9 [53].

## 5. Conclusions

In conclusion, the present study clearly demonstrates that luteolin-7-O-glucoside can significantly reduce the oral cancer metastasis by mitigating p38-induced increased expression of MMP-2. The study identifies luteolin-7-O-glucoside as a potential anti-cancer candidate that can be utilized clinically for improving oral cancer prognosis.

## Figures and Tables

**Figure 1 biomolecules-10-00502-f001:**
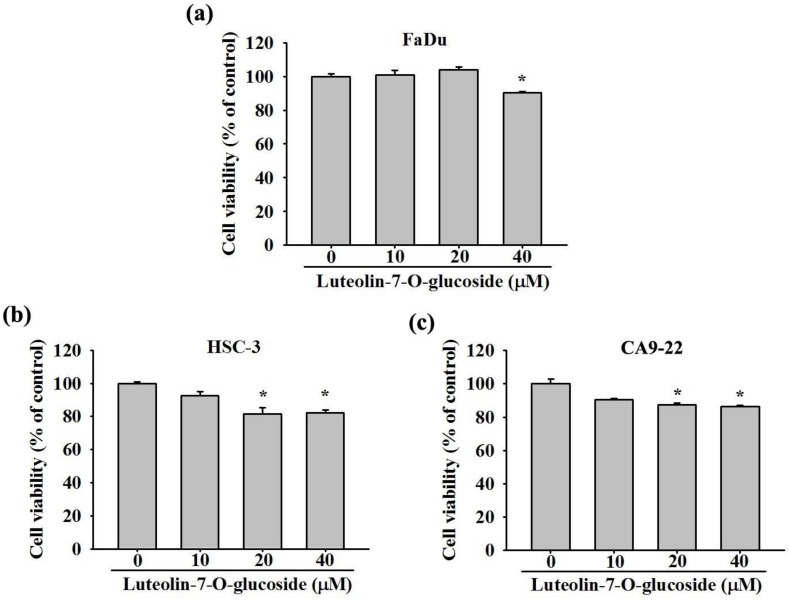
Cytotoxicity of Luteolin-7-O-glucosidein human oral cancer cells. (**a**) FaDu, (**b**) HSC-3, and (**c**) CA9-22 cell lines were treated with various concentrations (0, 10, 20, and 40 μM) of luteolin-7-O-glucoside for 24 h, and the cell viability was determined by MTT assay. The values are represented as mean ± SD of at least three independent experiments. **p* < 0.05, compared with the control group.

**Figure 2 biomolecules-10-00502-f002:**
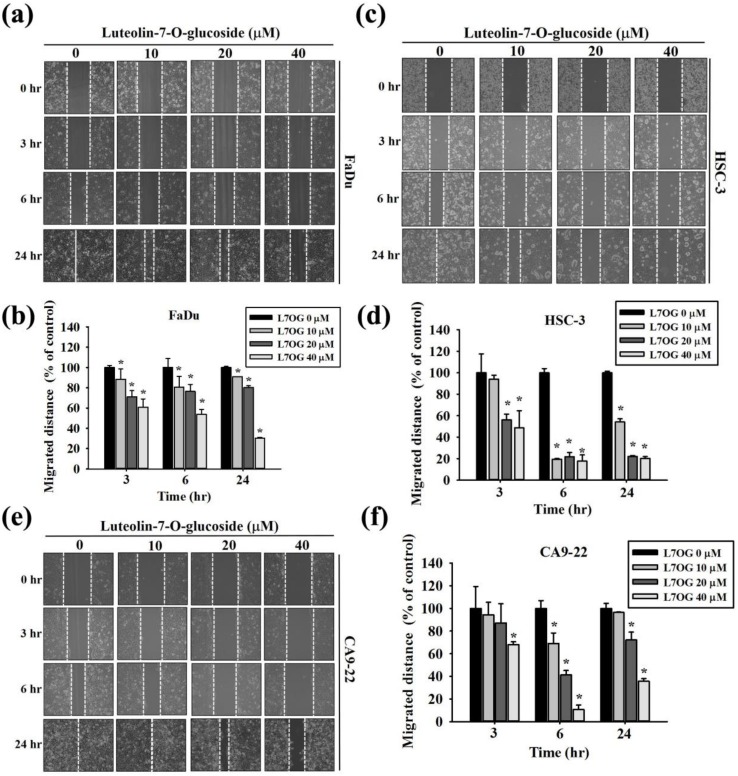
Luteolin-7-O-glucoside inhibits cell motility inhuman oral cancer cells. The effect of luteolin-7-O-glucoside treatment on cell motility was analyzed in (**a**,**b**) FaDu, (**c**,**d**) HSC-3, and (**e**,**f**) CA9-22 cells. The values are represented as mean ± SD of at least three independent experiments. **p* < 0.05, compared with the control group.

**Figure 3 biomolecules-10-00502-f003:**
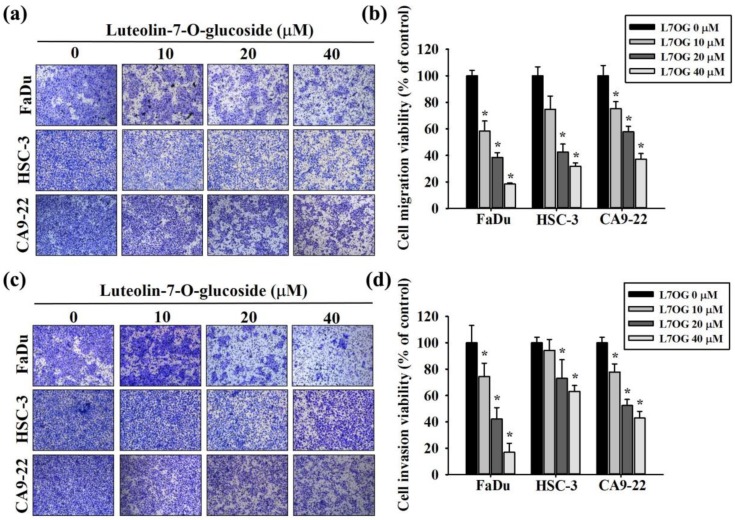
Luteolin-7-O-glucoside inhibits cell migration and invasion inhuman oral cancer cells. The effect of luteolin-7-O-glucoside treatment on cell migration (**a**) and invasion (**c**) was measured using trans-well assay in FaDu, HSC-3, and CA9-22 cells. The percentages of cells in migration and invasion assays are shown in (**b**) and (**d**), respectively. The values are represented as mean ± SD of at least three independent experiments. **p* < 0.05, compared with the control group.

**Figure 4 biomolecules-10-00502-f004:**
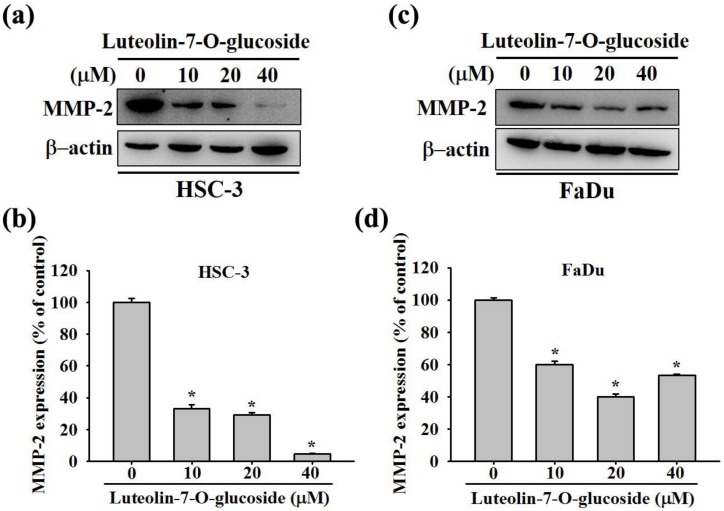
Luteolin-7-O-glucoside reduces the protein expression of matrix metalloproteinase (MMP)-2 in human oral cancer cells. The protein expression of MMP-2 was determined using Western blot in (**a**) HSC-3 and (**c**) FaDu cells. The quantitative results are shown in (**b**) and (**d**). The values are represented as mean± SD of at least three independent experiments. **p* < 0.05, compared with the control group.

**Figure 5 biomolecules-10-00502-f005:**
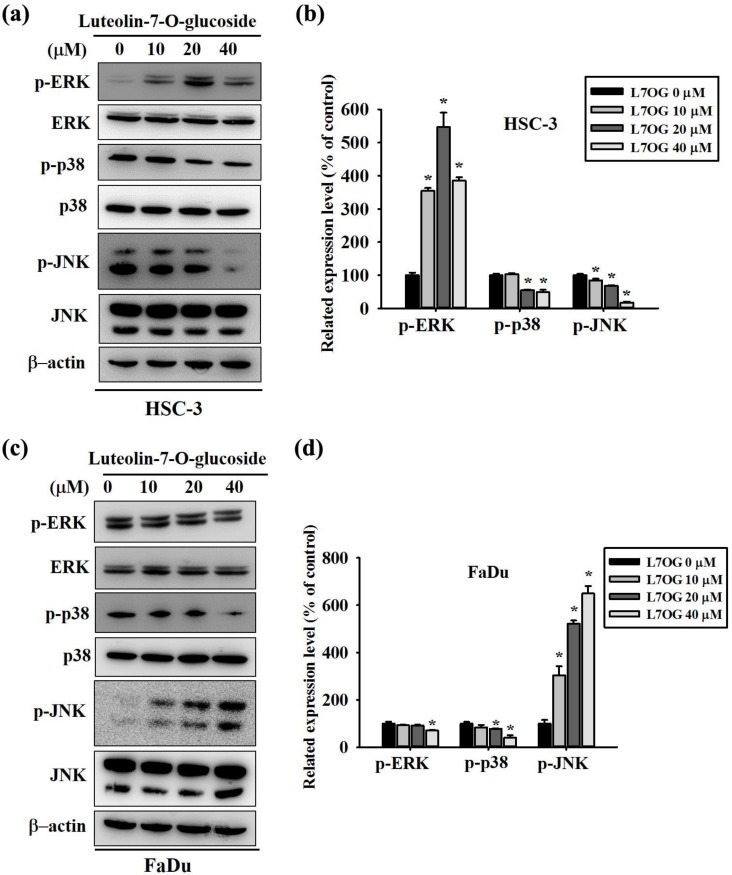
Luteolin-7-O-glucoside inhibits p38 pathway in human oral cancer cells. The phosphorylation as well as the total protein expressions of extracellular signal-regulated kinase 1/2 (ERK1/2), Jun N-terminal kinase 1/2 (JNK1/2), and p38 were measured after luteolin-7-O-glucoside treatment for 24 h in (**a**,**b**) HSC-3 and (**c**,**d**) FaDu cell lines. The values are represented as mean ± SD of at least three independent experiments. **p* < 0.05, compared with the control group.

**Figure 6 biomolecules-10-00502-f006:**
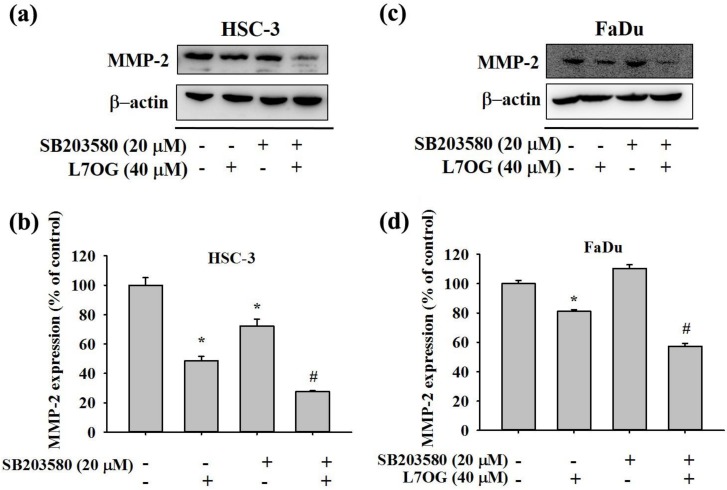
Effect of SB203580 and luteolin-7-O-glucoside co-treatment on MMP-2 protein expression in HSC-3 and FaDu cell lines. (**a**,**b**) HSC-3 and (**c**,**d**) FaDu cell lines were pre-treated with SB203580 for 1h, followed by treatment with Luteolin-7-O-glucoside for 24 h. Next, the culture medium was subjected to western blot assay to determine the MMP-2 expression. The values are represented as mean ± SD of at least three independent experiments. **p* < 0.05, compared to the control group; #*p* < 0.05, compared to the luteolin-7-O-glucoside treated group.

**Figure 7 biomolecules-10-00502-f007:**
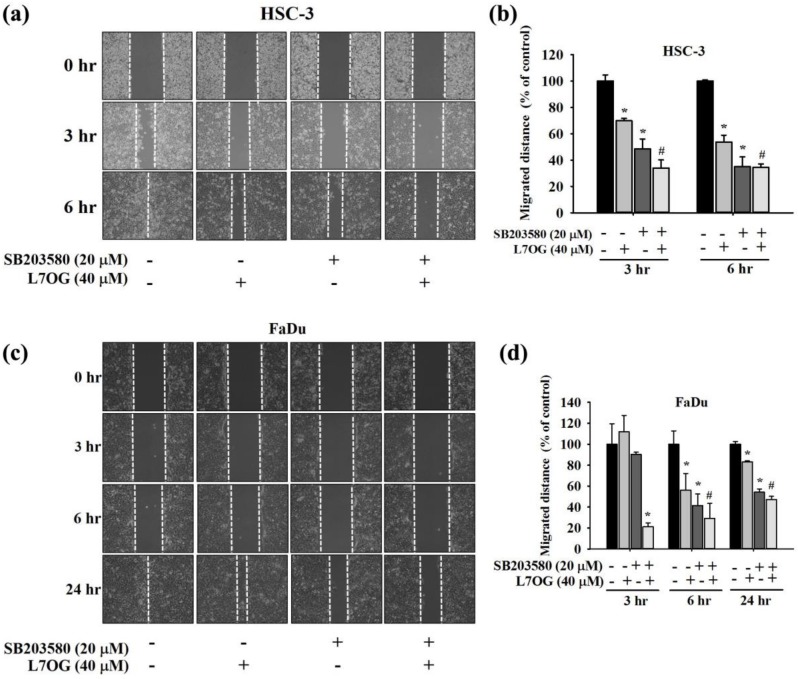
Effect of SB203580 and luteolin-7-O-glucoside co-treatment on cell motility in HSC-3 and FaDu cell lines. The cell motility was measured using wound healing assay after the co-treatment with SB203580 and luteolin-7-O-glucosidein (**a**,**b**) HSC-3 and (**c**,**d**) FaDu cell lines. The values are represented as mean ± SD of at least three independent experiments. **p* < 0.05, compared to the control group; #*p* < 0.05, compared to the luteolin-7-O-glucoside treated group.

**Figure 8 biomolecules-10-00502-f008:**
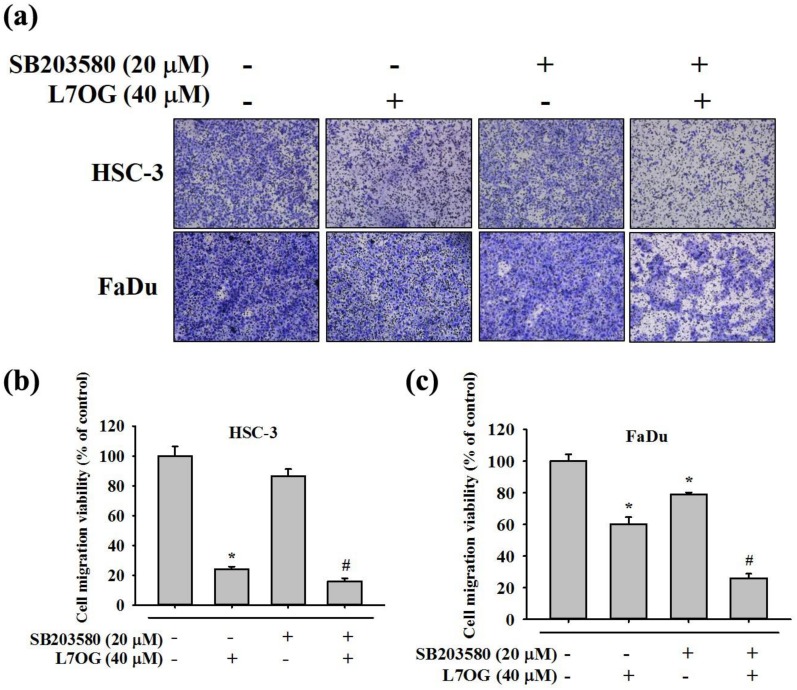
Effect of SB203580 and luteolin-7-O-glucoside co-treatment on cell migration in HSC-3 and FaDu cell lines. The cell migration (**a**) was measured using trans-well assay after the co-treatment with SB203580 and luteolin-7-O-glucosidein HSC-3 and FaDu cell lines. The quantitative results are shown for (**b**) HSC-3 and (**c**) FaDu cells. The values are represented as mean ± SD of at least three independent experiments. **p* < 0.05, compared to the control group; #*p* < 0.05, compared to the luteolin-7-O-glucoside treated group.

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
