# Peer review of "Luteolin-7-O-Glucoside Inhibits Oral Cancer Cell Migration and Invasion by Regulating Matrix Metalloproteinase-2 Expression and Extracellular Signal-Regulated Kinase Pathway"

_biomolecules, 2020, doi:10.3390/biom10040502_

Round 1
Reviewer 1 Report
The authors study the effect of the natural plant flavonoid luteolin-7-O-glucoside (L7G) on squamous cell carcinoma cell lines in vitro. They report that L7G inhibits cells mobility and invasion in vitro and that this may be due to its down-regulatory effect on the expression of matrix metalloproteinase-2 protein. Upon analyzing the possible involvement of the MAP kinase pathway they find that L7G inhibits phosphorylation of the p38 kinase. Inhibition of p38 by the standard inhibitor SB 203580 also results in an inhibition of p38 phosphorylation, MMP2 expression and cell motility and this effect is greater when both components are given. This strengthens the argument that a p38 induced MMP-2 secretion is necessary for cell migration and invasion which can be inhibited by L7G.
The presented data are sound and partially novel. However as the authors point out themselves the effect of luteolin on cancer cell metastasis has been studied before in other cancer models.
One question that remains for me is: why are the authors using the glucoside form of luteolin. When given orally the glucoside will be cleaved off by enzymes of the digestive tract and only luteolin will be taken up by intestinal cells. So what strategy would be used to treat patients of this cancer type. Maybe the authors can elaborate a litte bit on this.
Also the effects reported would be more convincing if also primary tumor cells would have been included in the study. Cell lines are always somewhat artificial models as the authors report themselves by the very different effects of L7G on Erk and Jnk phosphorylation in the two cell lines used.
Minor points:
- In the introduction there are some incomplete sentences.
- the data for cell migration and invasion in Fig. 3 look very similar. Whats the difference in the two determinations?
- lanes 202 and 219: the authors do not study the protein "expression" of the kinases but their "phosphorylation status".
- For fig 5 the time point of cell lysis is missing
Author Response
Comments and Suggestions for Authors
The authors study the effect of the natural plant flavonoid luteolin-7-O-glucoside (L7G) on squamous cell carcinoma cell lines in vitro. They report that L7G inhibits cells mobility and invasion in vitro and that this may be due to its down-regulatory effect on the expression of matrix metalloproteinase-2 protein. Upon analyzing the possible involvement of the MAP kinase pathway, they find that L7G inhibits phosphorylation of the p38 kinase. Inhibition of p38 by the standard inhibitor SB 203580 also results in an inhibition of p38 phosphorylation, MMP2 expression and cell motility and this effect is greater when both components are given. This strengthens the argument that a p38 induced MMP-2 secretion is necessary for cell migration and invasion which can be inhibited by L7G.
The presented data are sound and partially novel. However, as the authors point out themselves the effect of luteolin on cancer cell metastasis has been studied before in other cancer models.
One question that remains for me is: why are the authors using the glucoside form of luteolin. When given orally the glucoside will be cleaved off by enzymes of the digestive tract and only luteolin will be taken up by intestinal cells. So, what strategy would be used to treat patients of this cancer type. Maybe the authors can elaborate a little bit on this.
Answers: Thanks for this valuable suggestion. The main reason is the relationship of luteolin-7-O-glucoside on oral cancer is still not clear. In addition, we also agree with the reviewer comment. Due to the glucoside will be cleaved off by enzymes of the digestive tract, we evaluate that if it can be applied to the clinical treatment of oral cancer in the future, oral sprays drugs can be made and evaluated by spraying into the oral cavity.
Also, the effects reported would be more convincing if also primary tumor cells would have been included in the study. Cell lines are always somewhat artificial models as the authors report themselves by the very different effects of L7G on Erk and Jnk phosphorylation in the two cell lines used.
Answers: Thanks for this valuable suggestion. Indeed, we very agree with this comment. If there are relative data to evaluate the molecular mechanism of luteolin-7-O-glucoside in primary tumor cells, the research results can have more evidences and credibility. We will pay attention to this opinion in our future experiment.
Minor points:
- In the introduction there are some incomplete sentences.
Answers: We apologize for these confuse. The follow sentence has been modified in revised manuscript.
Line 43-48: Oral squamous cell carcinoma that primarily affects the lips, oral cavity, and tongue represents about 90% of all head and neck cancers. Together with pharyngeal cancer, oral carcinoma has become the sixth most common type of cancer globally [1]. Despite advancement in diagnostic processes and therapeutic interventions, oral cancer is associated with a high rate of morbidity and mortality [2]. Because of the initial asymptomatic nature, oral cancer is often diagnosed in the later stage, resulting in distant metastasis and poor prognosis [3-6].
- The data for cell migration and invasion in Fig. 3 looks very similar. What’s the difference in the two determinations?
Answers: We apologize for these confuse. The differences between invasion and migration experimental methods is the cells must break down a layer of extracellular matrix gel coating in the experiment of invasion. We have modified the sentence in revised manuscript.
- lanes 202 and 219: the authors do not study the protein "expression" of the kinases but their "phosphorylation status".
Answers: We apologize for these confuse. The follow sentence has been modified in revised manuscript.
Line 187: Since the expressions and activities of MMPs are tightly regulated by MAPK pathway components, we next investigated the protein phosphorylation status of ERK, p38, and JNK in FaDu and HSC-3 cells after employing similar treatment as before.
Line 205: Since we observed a consistent change in p38 phosphorylation status in all the oral cancer cells
- For fig 5 the time point of cell lysis is missing
Answers: We apologize for these confuse. The follow sentence has been modified in revised manuscript.
Line 198: Figure 5. Luteolin-7-O-glucoside inhibits p38 pathway in human oral cancer cells. The phosphorylation as well as the total protein expressions of ERK1/2, JNK1/2, and p38 were measured after luteolin-7-O-glucoside treatment for 24 h in (A) (B) HSC-3 and (C) (D) FaDu cell lines. The values are represented as mean± SD of at least three independent experiments. *P<0.05, compared with the control group.
Reviewer 2 Report
Dear Editors,
dear colleagues,
the present manuscript of Velmurugan et al. details interesting data about luteolin-7-O-glucoside in regard to potential anti-tumor functions.
Although I feel that the manuscript should be publihed for your esteemed readership, I have some concerns, which should be adressed by the author(s).
1) In the material&methods section, the authors provide information about zymography. However, I could not find any results regarding this technique, which would be very helpful, because expression does not necessarily correlate to function.
2) Although the authors used a P38 inhibitor, they did not use ERK inhibitor(s) (for example FR 180204) and/or MMP2 inhibitor(s). Since selectivity and specificity of inhibitors are always problematic, the authors should at least detail their decision to use only the P38 inhibitor in the discussion section.
3) Some words about other MMPs in regard to tumor pregression should be provided in the discussion section. Also, some words about the putative lutelin receptor and possible functions in regard to tumor progression could improve the discussion.
4) Results detailed in Fig. 1: Did the authors check any other time points? Since reduction of overall growth could be due to apoptosis and/or reduced growth, did the authors consider to check with apoptosis/cell growth assays (PI based cell cycle analysis for example).
5) throughout the text I found several syntax / spelling mistakes. The authors should carefully proof-read the text.
Here some mistakes, I detected:
ll 44-45 sentence is incomplete.
l 59 separate "that the"
ll 68-73 check grammar
l 135 redundant description.
l 169 check headline for word separation
fig. 3 headline
headline 3.6. spelling O-glucoside
Best regards
Author Response
Comments and Suggestions for Authors
The present manuscript of Velmurugan et al. details interesting data about luteolin-7-O-glucoside in regard to potential anti-tumor functions. Although I feel that the manuscript should be published for your esteemed readership, I have some concerns, which should be addressed by the author(s).
- In the material methods section, the authors provide information about zymography. However, I could not find any results regarding this technique, which would be very helpful, because expression does not necessarily correlate to function.
Answers: We apologize for these confuse. The zymography sentence has been removed in revised manuscript.
- Although the authors used a P38 inhibitor, they did not use ERK inhibitor(s) (for example FR 180204) and/or MMP2 inhibitor(s). Since selectivity and specificity of inhibitors are always problematic, the authors should at least detail their decision to use only the P38 inhibitor in the discussion section.
Answers: Thanks for this valuable suggestion. The follow sentence has been added to discussion section in revised manuscript.
Line 291: According to the results in figure 5, luteolin-7-O-glucoside treatment caused a significantly reduction in p38 and JNK phosphorylation in HSC-3 cells. In case of FADU cells, the treatment caused similar reduction in p38 phosphorylation as observed in HSC-3 cells (Figure 5c and d).
- Some words about other MMPs in regard to tumor progression should be provided in the discussion section. Also, some words about the putative lutelin receptor and possible functions in regard to tumor progression could improve the discussion.
Answers: Thanks for this valuable suggestion. The follow sentence has been added to discussion section in revised manuscript.
Line 287: Luteolin suppress vascular endothelial growth factor receptor (VEGF) 2 and mediated human prostate tumor growth and angiogenesis [42].
Line 302: Several MMPs such as MT1‐MMP, MMP‐2 and MMP‐9 were relation to the stages of cancer cell invasion progression.
- Results detailed in Fig. 1: Did the authors check any other time points? Since reduction of overall growth could be due to apoptosis and/or reduced growth, did the authors consider to check with apoptosis/cell growth assays (PI based cell cycle analysis for example).
Answers: Thanks for this valuable suggestion. At the beginning of the study, we have analyzed different time points (24, 48, and 72 hours) in figure 1. However, according to the results, the cell survival was significantly inhibited after 48 hours. We have already further research and analysis, and related results will be published in the future.
- Throughout the text I found several syntax / spelling mistakes. The authors should carefully proof-read the text. Here some mistakes, I detected:
44-45 sentence is incomplete.
Answer: We apologize for these confuse. The follow sentence has been modified in revised manuscript.
Line 43-48: Oral squamous cell carcinoma that primarily affects the lips, oral cavity, and tongue represents about 90% of all head and neck cancers. Together with pharyngeal cancer, oral carcinoma has become the sixth most common type of cancer globally [1]. Despite advancement in diagnostic processes and therapeutic interventions, oral cancer is associated with a high rate of morbidity and mortality [2]. Because of the initial asymptomatic nature, oral cancer is often diagnosed in the later stage, resulting in distant metastasis and poor prognosis [3-6].
59 separate "that the"
Answer: We apologize for these confuse. Those mistakes have been modified in revised manuscript.
68-73 check grammar
Answer: We apologize for these confuse. The follow sentence has been modified in revised manuscript.
Line 66-72: Moreover, previous studies have shown that human lung cancer cell migration and invasion can be suppressed by pharmacologically downregulated ERK/p38 signaling pathway and inhibited MMP-2 and MMP-9 activities [21]. Similarly, in ovarian cancer, increased cell growth and migration by Rap1A, a Ras-associated protein, has been shown to be associated with elevated ERK/p38 and notch signaling [22]. Taken together, it is well-evidenced that the complex crosstalk between MAPK signaling pathway components and MMPs plays an immensely important role in regulating cancer metastasis and progression.
135 redundant description.
Answer: Thanks for this valuable suggestion. The redundant description has been removed in revised manuscript.
169 check headlines for word separation
Answer: We apologize for these confuse. Those mistakes have been modified in revised manuscript.
Round 2
Reviewer 2 Report
Dear editor, dear colleagues,
the present version of the manuscript should be considered for publication.
The authors answered all questions sufficiently.
However, I think I still found few mistakes/remarks and suggest following changes:
l53 MMps are a target for...
L59 introduce a gap between ...p38 causes….
l262 ...has shown that lutelin...
l302 ….were found correlated….
Best regards
Author Response
The present version of the manuscript should be considered for publication.
The authors answered all questions sufficiently.
However, I think I still found few mistakes/remarks and suggest following changes:
Answers: Thanks for this valuable suggestion. The follow sentence has been modified in revised manuscript.
L53 MMps are a target for...
Line 53: MMPs are a target for developing treatment strategies against cancer. In this context, several studies have shown that selective inhibition of MMP can be associated with better cancer management [9-11].
L59 introduce a gap between ...p38 causes….
Line 59: Introduce a gap between MAPK signaling pathway and MMP activity, it is known that the stimulation of p38 causes increased metastasis of neoplastic squamous epithelial cells by regulating the expression of MMPs [13].
L262 ...has shown that lutelin...
Line 262: Of these doses, 20 and 40 μM has shown that luteolin-7-O-glucoside significant anti-proliferative or cytotoxic effects (Figure 1).
l302 ….were found correlated….
Line 302: Several MMPs such as MT1‐MMP, MMP‐2 and MMP‐9 were found correlated to the stages of cancer cell invasion progression.